# Demographics, Diagnoses, Drugs, and Adjuvants in Patients on Chronic Opioid Therapy vs. Intermittent Use in a Tertiary Pediatric Chronic Pain Clinic

**DOI:** 10.3390/children10010037

**Published:** 2022-12-24

**Authors:** James A. Tolley, Martha A. Michel, Elisa J. Sarmiento

**Affiliations:** 1Section of Pediatric Anesthesia, Department of Pediatrics, Riley Hospital for Children, Indiana University School of Medicine, Indianapolis, IN 46202, USA; 2Department of Biostatistics and Health Data Science, Indiana University School of Medicine and Richard M. Fairbanks School of Public Health, Indianapolis, IN 46202, USA

**Keywords:** opioids, pediatric, pain, pediatric pain, chronic pain, pain treatments

## Abstract

Anywhere from 11.6% to 20% of pediatric and adolescent patients treated for chronic pain are prescribed opioids, but little is known about these patients. The purpose of this study was to determine the characteristics of patients on chronic opioid therapy (COT) and what therapies had been utilized prior to or in conjunction with COT. The study was a retrospective chart review of all chronic pain patients seen during 2020 with those patients on COT separated for analysis. A total of 346 unique patients were seen of which 257 were female (74.3%). The average age was 15.5 years. A total of 48 patients (13.9%) were identified as being on COT with an average age of 18.1 years. Of these, 23 (47.9%) were male which was significantly more than expected. The most common reason for patients to be receiving COT was palliative (13/48), and the second most common was sickle cell anemia (10/48). Patients on COT were significantly more likely to be male, be older, and to be concurrently prescribed benzodiazepines. Concurrent opioid and benzodiazepine therapy is a risk factor for respiratory depression and overdose. Further investigation into the increased proportion of males and benzodiazepine usage in patients on COT is warranted.

## 1. Introduction

In the mid-1990s, the medical community began to change its focus regarding the patient experience of pain, both acute and chronic [1,2]. The VA published a toolkit outlining a nationwide initiative for pain management using pain as the fifth vital sign [3], and Congress dedicated a decade to research into pain and its management starting in the year 2000 [4,5]. This emphasis on pain meant that chronic pain was no longer simply a symptom, but a neurologic disease entity [6], with the biopsychosocial model becoming increasingly prevalent [7]. A multidisciplinary approach to the treatment of chronic pain was to become the de-facto the standard of care [7,8] due to evidence demonstrating its benefit [8].

However, this focus on identifying and treating pain had some unintended negative consequences. The number of opioid prescriptions in the United States increased dramatically during the decade of pain peaking in 2012 nationally with morphine milligram equivalents peaking in 2010 [9,10]. A similar overall pattern occurred in pediatric and young adult patients with steady decreases since 2013 [11]. Despite this decreasing trend in prescriptions, the opioid epidemic rages on and the number of deaths from opioid overdose continues to increase [12]. The extent of the contribution of prescription opioid use to the opioid crisis remains controversial [13]. A longitudinal study of high school seniors found that there was a pattern of medical use of opioids prior to nonmedical use [14], but that proper medical use did not lead to substance use disorder (SUD), whereas nonmedical use could predict later development of SUD [15]. A survey of opioid use and misuse which included adolescents 12 to 17 years of age found that 3.8% had misused opioids and that 25.4% of the time, they had gotten those opioids through the healthcare system [16]. 

Despite the recognition of the value of multidisciplinary treatment in managing acute and chronic pain in pediatric patients [17], 11.6% to 20% of pediatric patients with chronic pain were prescribed opioids either at initial presentation to a tertiary care chronic pain clinic or through identification via a prescription claims data [18,19,20]. Little is known, however, about the characteristics of patients receiving ongoing chronic opioid therapy (COT) in the pediatric tertiary care chronic pain clinic. Therefore, the purpose of this study was to identify which patients received COT and the conditions being treated, what non-opioid medications were employed, and what other modalities were utilized as part of a multidisciplinary approach to treating pediatric chronic pain to search for potential areas for focus and improvement to minimize the use of opioids given the potential for misuse of prescriptions [16].

## 2. Materials and Methods

The study was considered exempt from review by the Indiana University School of Medicine Institutional Review Board as it did not involve human subjects and was a review of medical records. Therefore, individual patient consent was not obtained.

The Riley Chronic Pain Center is a multidisciplinary tertiary pediatric pain clinic serving the entire state of Indiana with some patients also being referred from east central Illinois. Patients may be referred to the clinic by their primary care provider or by multiple specialists outside or within the IU Health system. We receive referrals from our rheumatologic, neurological, orthopedic, and neurosurgical colleagues frequently. Hematology will refer patients expected to require or currently requiring COT in the management of pain associated with sickle cell anemia. 

The scheduling module from the electronic medical record (EMR) platform as provided by Cerner was used to identify all patients with a chronic pain clinic appointment, whether in person or through a virtual platform, during the calendar year 2020. All patients who attended an appointment and were evaluated by a physician or advanced practice provider were included. No patients who had a completed provider visit note were excluded from the analysis.

The patient’s date of birth, medical record number, gender at birth, and diagnosis prompting chronic pain clinic evaluation for each individual patient were recorded in an Excel spreadsheet that was maintained in a database on an encrypted and secured computer. The average age of each patient for select groups was determined by using 1 July 2020, since the age would vary from the beginning to the end of the year.

The EMR was then reviewed to determine if the patient had been prescribed an opioid during 2020. All patients prescribed opioids regardless of the dosage were separated from those patients who had not been prescribed an opioid at any time during 2020. Patients were then separated into those receiving COT and those receiving isolated or intermittent opioid (IO) prescriptions based upon the clinical judgement of the primary author with agreement from the secondary authors until consensus was reached. As it turns out, 42 of the 48 patients on COT were managed by the primary author and 43 of the 48 patients received monthly opioid prescriptions throughout 2020 so consensus was easy to achieve. Morphine milligram equivalents (MME) per prescription, MME total for the year, and average daily MME for each patient were also calculated based upon data provided by the state’s prescription drug monitoring program which is incorporated into the EMR. Any prescriptions for opioids by any provider in the state would have been captured and included in this manner. The diagnosis for each patient receiving COT was also recorded. 

The EMR for each patient receiving an opioid prescription in 2020 was then reviewed to determine if the patient had either tried or was currently utilizing certain pharmacologic or nonpharmacologic treatments. Pharmacologic treatments included such drug classes as anticonvulsants, antidepressants, non-steroidal anti-inflammatory medications including acetaminophen (NSAIDs), clonidine, muscle relaxants including diazepam, other medications such as ketamine or cannabidiol (CBD), and all topical agents regardless of medication class. Nonpharmacologic treatments were grouped into physical therapy including aqua and land-based therapy; regional or neuraxial nerve blocks; other modalities (such as acupuncture, massage, or intravenous fluid therapy); and finally psychologic modalities, such as cognitive behavioral therapy (CBT), biofeedback, and hypnosis.

Analysis was performed using the Chi-square or Fisher’s Exact test to estimate differences in proportions between groups, while for continuous variables, the T-test was used. All statistical analyses were performed using SAS Version 9.4 (SAS Institute, Cary, NC, USA).

## 3. Results

A total of 346 unique patients were seen and evaluated in 2020, of whom 257 (74.3%) were female and 89 (25.7%) were male. The mean age for all chronic pain patients was 15.5 years. Of those 346 patients, 74 (21.4%) had received an opioid prescription at some point during the year. These 74 patients had an average age of 17.2 years and 48 (58.1%) were female. The number of patients identified as being on COT was 48, representing 13.9% of the total individuals treated in the chronic pain clinic in 2020. Of the 48 patients who were identified as receiving COT, 25 (52.1%) were female and 23 (47.9%) were male. The data and comparisons between patients not prescribed opioids and all patients prescribed opioids, both COT and IO users, are summarized in Table 1.

The average daily MME for the 26 IO patients was 10.1 with a range of 1.25 to 28.8, and the total MME for the year in these patients ranged from a low of 15 to a high of 345 with an average of 120.9. For those 48 patients on COT, the average daily MME was 81.6 with a range of 6.7 to 596.2, while the total MME for the year in these patients ranged from 80 to 4160 with an average of 979. Of these patients, 9 were calculated as having received greater than 90 MME per day over the course of 2020.

The most common Indication for COT was of a palliative nature, e.g., wheelchair- bound or ventilator-dependent individuals with varying sources of pain. The second most common indication for COT was for those patients with severe pain associated with sickle cell anemia. The complete list of diagnoses for which patients were receiving COT along with the average number of treatment modalities employed per diagnosis can be seen in Table 2. Patients with the highest average number of total treatment modalities tried or currently utilized was the single patient with chronic pancreatitis, the 3 patients with complex regional pain syndrome, and the 7 patients with fibromyalgia.

Of the 48 patients receiving COT, 47 (97.9%) were either taking or had received an anticonvulsant, 43 (89.6%) were either taking or had received an NSAID, and 39 (81.3%) were either taking or had received an antidepressant. The remainder of the treatment modalities, both pharmacologic and nonpharmacologic, that were either being utilized or had been tried at some point in the treatment of chronic pain can be found in Table 3 for both those on COT and those receiving intermittent opioids (IO).

The only statistically significant difference in therapies between the 26 patients receiving intermittent opioids and the 48 patients on COT was the proportion receiving concurrent benzodiazepines with a *p*-value of 0.0378. The proportion of all other therapies either tried or currently being utilized between the two groups failed to reach statistical significance. Concurrent benzodiazepine prescriptions occurred in 10 of the 13 palliative patients.

Intermittent opioid patients had tried or were using an average of 4.46 pharmacologic therapies, whereas COT patients had tried or were using an average of 4.5 pharmacologic therapies. Nonpharmacologic therapy usage was 1.38 in IO patients and 1.48 in COT patients on average. Neither of these were statistically different.

## 4. Discussion

The purpose of the present study was to identify the characteristics of patients prescribed COT, detail the therapies utilized, and identify any areas for possible improvement in the management of complex chronic pain patients in a multidisciplinary chronic pediatric pain clinic. We found that a total of 346 patients had been seen during the year 2020 and that 74 (21.4%) had received an opioid prescription during the year. Some had weaned off methadone during the year or had received a single prescription for a procedure or wisdom teeth removal.

We found that 13.9% (48/346) were receiving COT which is consistent with the 11.6% to 20% found in the literature [18,19,20]; however, in the study by Richardson et al. [18], the 11.6% of patients on opioids were already on opioids at presentation. The demographics of our patients are also consistent with other clinics as reported in the literature. In the Richardson et al. [18] study, the mean age at presentation was 14.5 years and 71.26% were female. Vetter [21] reported a mean age of 14.0 years and 73% female for the first 100 patients when our clinic was started from May 2005 through October 2007. Our mean age in this current sample of 346 patients is 15.5 years and 74.3% female. We did not limit our analysis to new patients only and so our average age is a little higher as many of our patients had been treated in the clinic for longer than one year.

The retrospective database review by Gmuca et al. [20] used health claims data to evaluate patients between the ages of 2 and 18 with a diagnosis of unspecified myalgia and myositis and found that 54.5% were female and 19.4% had received an opioid prescription. They found that female sex was associated with opioid exposure, whereas we noted that the proportion of males receiving COT was significantly more than would be expected based upon our patient demographics. Like Richardson et al. [18], we found that those receiving opioids were older, and this difference was statistically significant.

We originally sought to define COT using daily MME. This was not practical as some patients had transitioned to an adult clinic during the year, and another had moved out of state so losing patients to follow-up early in the year would yield a lower-than-expected value for daily MME averaged over a year. The average daily MME for those on COT was 81.6 with range from 6.7 to 596.2. It might be that the total MME for the year would provide a better assessment of COT considering the range of 15 to 345 for non-COT patients and 80 to 4160 for COT patients. There is still some overlap of values since some COT patients had not started COT until later in the year, and some were no longer part of the pain clinic early in the year. Indeed, defining long-term opioid therapy is anything but straightforward as evidenced by this review which found 41 definitions in 34 studies [22]. 

Nine of the 46 patients (19.6%) on COT were prescribed greater than 90 MME/day. The 2016 CDC guidelines for prescribing opioids for chronic pain recommend avoiding dosages above 90 MME/day without careful justification [23]. New guidelines released in November 2022 may provide more flexibility in prescribing opioids [24]. These guidelines do not address using opioids in the management of patients with sickle cell anemia, malignancy, or as part of palliative care therapy which accounted for 54.3% of our patients on COT. Nor do they specifically address opioid prescribing in pediatric patients.

A large percentage of both groups of our patients were taking or had trialed gabapentinoids, antidepressants, or NSAIDs. Donado et al. [25] found that, in their institution between 2013 and 2019, gabapentin prescriptions had increased 1.4-fold and pregabalin 1.3-fold. In the chronic pain clinic at that institution, they found that 24% of 650 patients presenting for their first visit had been previously treated with gabapentin, 1% with pregabalin, and 3% with both. Of those, 44% had stopped due to mild or moderate adverse effects or lack of efficacy. An additional 120 patients received a gabapentinoid prescription at that first visit.

There is little data on efficacy of gabapentinoids in the treatment of pediatric chronic pain as stated in the systematic review by Egunsola et al. [26], although a retrospective analysis of 22 nonverbal children showed a significant decrease in pain behaviors in 21 of them [27].

Likewise, there is little data to support the use of antidepressants in pediatric and adolescent chronic pain management. A Cochrane review published in 2017 [28] found only a small number of studies for review with insufficient data to perform a meta-analysis. In 2019, a placebo-controlled trial of duloxetine in adolescents with fibromyalgia over 13 weeks found that more patients on duloxetine had a reduction in pain severity even though the primary endpoint was not met [29]. A meta-analysis by Jolly et al. published in 2021 [30] found that psychotropic medications did demonstrate analgesic efficacy, however, the number of included studies was small. Drugs evaluated included amitriptyline, citalopram, duloxetine, and buspirone.

NSAIDs were also used in 67 of the 72 patients (93%) in our clinic who received an opioid prescription in 2020 despite little evidence as to their efficacy. A 2017 Cochrane review found seven studies none of which compared the NSAID in question to a placebo [31]. The authors concluded that there was insufficient data for analysis. Many of the same authors were unable to find studies for inclusion in a systematic review of paracetamol (acetaminophen) [32], although they did acknowledge that there are some adult randomized controlled trials showing benefit with both acetaminophen [32] and some NSAIDs [31]. NSAIDs have also been found to have lower abuse potential (2.5%) than tramadol (2.7%) or hydrocodone (4.9%) over a 12-month period [33]. 

In contrast to the above pharmacologic treatments, nonpharmacologic therapies, such as cognitive behavioral therapy (CBT), have been shown to provide significant benefits in relation to pain and functioning for various pediatric chronic pain disorders. In a randomized trial of 48 children aged 11–17 years with chronic headache, abdominal pain or musculoskeletal pain, Palermo et al. [34] demonstrated that internet delivered family CBT could significantly reduce pain intensity and improve functioning following an 8-week program which was sustained at the 3-month follow-up. Acceptance and Commitment Therapy [35] has also been shown in a randomized trial to improve pain intensity and functional ability for up to 6.5 months of follow-up in a wide range of painful conditions [36]. An established program of CBT for youth with sickle cell anemia can decrease pain, improve function, and decrease healthcare utilization [37].

A neuromuscular exercise training pilot program for teenagers with fibromyalgia found that an 8-week exercise program provided significant additional pain improvement at the 3-month follow-up compared to CBT alone [38]. The type of exercise may play a role in outcomes as demonstrated by a randomized trial in 50 children with juvenile idiopathic arthritis [39]. Pilates was more effective in reducing pain and improving quality of life compared to a conventional exercise program over a period of six months. In a retrospective review, Leonard et al. [40] found that aquatic therapy performed during hospitalization for painful crises in patients with sickle cell anemia increased time between hospitalizations by 26% although hospital stays were longer possibly due to the availability of aquatic therapy.

Despite some promising studies demonstrating the effectiveness of certain nonpharmacologic therapies for certain conditions, our patients who received opioids used these methods less often than anticonvulsants, NSAIDs, and antidepressants for which there is little data supporting efficacy. This may be due to the variety of reasons our patients are experiencing chronic pain. Some of our palliative patients are wheelchair bound and non-verbal which might preclude the use of CBT or physical therapy. This is one of the limitations of our study in that no patients were excluded from analysis. However, this does represent real world conditions, and despite this limitation, may provide some insight into patients who might benefit from nonpharmacologic therapies, such as aquatic therapy in our cohort of patients with sickle cell.

One of the barriers for increasing nonpharmacologic therapies in our patients is insurance coverage. Studies have shown marked differences in the coverage of physical therapy, CBT, massage, acupuncture and other treatments based upon the type of insurance coverage or the state insurance program [41,42]. Even if covered, there may be limitations on the number of treatments [42] causing increased reliance on medications, including opioids, for chronic pain conditions.

In an editorial discussing the implications of the 2016 CDC opioid guidelines [23] on children, Schechter and Waldo advocated a balanced approach taking into consideration the risks of opioids and the consequences of undertreating pain [43]. This is the approach we have tried to take in our chronic pain clinic as demonstrated in the present study.

It should be noted that this data is from the year 2020 which included the beginning days of the COVID pandemic. In our institution it took a little more than a week to have a system in place to conduct virtual telemedicine visits for both new and return patients after closing all clinics in mid-March. Within about 6 weeks, however, clinics had opened to in-person appointments with virtual appointments remaining available at the discretion of the patient or provider. For the COT patients in 2020, there were 71 in-person visits and 69 virtual appointments which did not influence their pharmacologic management at all. The pandemic undoubtedly had an impact on the availability of nonpharmacologic therapies, but since the majority (43 of 48) were already on opioids beginning in January 2020, this would not have helped prevent initiation of opioids.

Another limitation of the current study is the retrospective nature of the study which relies somewhat on the documentation of the provider during the initial and subsequent visits as it relates to the review of the medical record as well the potential recall bias of the patient and caregiver regarding the treatments tried previously and their effectiveness.

Rather than simply reporting the percentage of patients on opioids in a pediatric chronic pain clinic, the current study adds to existing knowledge by documenting the other pharmacologic and nonpharmacologic modalities that are also utilized in patients with chronic pain. Anticonvulsants and NSAIDs were used in over 90% of patients and antidepressants were used over 80% of the time. This demonstrates the desire to utilize opioids almost as a treatment of last resort which seems reasonable in the current environment.

One surprising result was the number of patients on COT and receiving concurrent benzodiazepines. This was undoubtedly due to the palliative nature of some of our patients, but there were patients with other diagnoses that also received concurrent benzodiazepines. Increased awareness in the COT population is important considering the increased risk of severe respiratory depression and mortality [44]. It would also be prudent to verify that each patient on COT has access to naloxone nasal spray and is instructed in its usage.

Some of the diagnoses for which COT were prescribed are controversial such as fibromyalgia (FM) and complex regional pain syndrome (CRPS). Opiates, except for tramadol, are not part of the treatment recommendations for FM [45] since evidence supporting their efficacy is lacking [46]. Despite this, 17% to 20% of pediatric patients with musculoskeletal pain including FM are prescribed opioids [19,20], and 27.1% of adults were taking opioids upon admission to a 2-day Fibromyalgia Treatment Program [47]. Our clinic saw 84 patients with FM in 2020 and 7 (8.3%) were on COT. This is less than what is reported in the literature. Additionally, these patients had tried or were currently using 7.4 other modalities for treatment of their pain which would suggest that patients were most severely affected by their symptoms and had markedly impaired functioning.

The 3 patients with CRPS had tried or were utilizing 10 different modalities on average to try to cope with their pain. We saw 25 patients with CRPS so 12% of our patients with CRPS were on COT. This compares to 19.3% of adolescent patients presenting to an inpatient rehabilitation program [48], although all patients were weaned of opioids prior to completion of the program. Improved access to such multidisciplinary rehabilitation programs would have the potential to decrease opioid usage among CRPS patients.

## 5. Conclusions

We found that 13.9% of our active chronic pain clinic patients were on COT during 2020 and that they were more likely to be older and male than those not on COT. The most common indication for COT was palliative in nature followed by patients with sickle cell anemia. Patients on COT were likely to be taking or have tried anticonvulsants, NSAIDs, and antidepressants. Concurrent benzodiazepine usage in our population of COT patients is a concern. There may be opportunities to improve the usage of nonpharmacologic therapies such as CBT and aquatic therapy in certain subsets of our patients. More research is needed to determine efficacious therapies, both pharmacologic and nonpharmacologic, in pediatric patients with chronic pain.

## Figures and Tables

**Table 1 children-10-00037-t001:** Demographics of patients.

Opioid Prescribed		No	Yes	*p*-Value
Gender, N (%)	Female	214 (76.7)	43 (58.1)	0.0003
	Male	58 (21.3)	31 (41.9)	
Age, Mean (SD)		15.01 (2.93)	17.21 (4.22)	<0.0001

**Table 2 children-10-00037-t002:** Indication for chronic opioid therapy (COT).

Diagnosis	Number out of 48	Percentage	Average Number of Treatment Modalities
Palliative Indication	13	27.1	6.8
Sickle Cell Anemia	10	20.8	5.7
Fibromyalgia	7	14.6	7.4
Neuropathic Pain	6	12.5	6.2
Musculoskeletal Pain	4	8.3	6.5
Complex Regional Pain Syndrome	3	6.3	10
Vascular Disorder	2	4.2	6
Malignancy	2	4.2	6.5
Chronic Pancreatitis	1	2.1	10

**Table 3 children-10-00037-t003:** Treatments for patients on chronic opioid therapy (COT) and intermittent opioids (IO).

Treatment	N of COT pts (%)	N of IO pts (%)	*p*-Value
Anticonvulsants	47 (97.9)	24 (92.3)	0.2808
Antidepressants	39 (81.3)	22 (84.6)	1.0000
NSAIDs (includes acetaminophen)	43 (89.6)	24 (92.3)	1.0000
Clonidine	19 (39.6)	13 (50.0)	0.3879
Muscle relaxants (includes diazepam)	29 (60.4)	13 (50.0)	0.4641
Other (ketamine, CBD)	15 (31.3)	3 (11.5)	0.0881
Topical medications of all classes	24 (50.0)	17 (65.4)	0.2037
Physical, aqua, occupational therapy	31 (64.6)	17 (65.4)	0.9450
Regional or neuraxial blocks	12 (25.0)	6 (23.1)	0.8540
Acupuncture, massage, intravenous fluids	10 (20.8)	4 (15.4)	0.7581
CBT, biofeedback, hypnosis	30 (62.5)	15 (41.7)	0.6859
Concurrent benzodiazepines	19 (39.6)	4 (15.4)	0.0378

## Data Availability

Data is available upon request from the corresponding author. Data are not publicly available due to privacy concerns related to the medical records of treated individuals.

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
