# Peer review of "Demographics, Diagnoses, Drugs, and Adjuvants in Patients on Chronic Opioid Therapy vs. Intermittent Use in a Tertiary Pediatric Chronic Pain Clinic"

_children, 2022, doi:10.3390/children10010037_

Round 1
Reviewer 1 Report
This is a well written retrospective review of patients presenting to a moderate-sized tertiary referral pain clinic with particular interest in contrasting those on chronic opioid therapy (COT) with those not on COT. The results are well described and consistent with literature and current practice. The discussion is appropriate but some points could be expanded/clarified. I had a few suggestions for the authors:
1. It would be helpful to describe the process for referral to your pain clinic as some programs may have robust palliative care or hematology programs that provide similar expertise.
2. Since this data is from the start of the covid pandemic, you might clarify how that impacted your referrals and patient access to non-pharmacologic therapies.
3. I appreciate that COT can be difficult to assess from medical or prescription records-please more clearly describe how this treatment category was operationalize for your data collection.
4. Was the yearly total MME obtained just from EHRs or was the data obtained/confirmed from state prescription drug monitoring databases, which might include prescriptions obtained outside of the clinic?
5. Given that you had access to EHR data, I would have thought that you could have associated benzodiazepines usage and diagnoses and I would have assumed that it would be largely restricted to the palliative care population-please clarify?
Author Response
Response to Reviewer Number 1
Thank you for the kind words and suggestions. I am responding to your comments and suggestions by number as you list them.
- I included more about the referral process, see lines 64-70 which is an additional paragraph.
- I added a paragraph in the discussion lines 279-289 stating the limitation of access to some therapies, but it wouldn’t have changed opioid prescriptions.
- 42 of the 48 patients were my patients and 43 of the 48 patients received essentially monthly prescriptions (see point 2 above). It was very easy to figure out who was COT vs who was not. There were only a few patients that started COT later in the year, but it was really easy to get consensus between myself and the second author. I mention the above in the methods section also line 89-91.
- I included information that the PDMP data is incorporated into the EMR and included all opioid prescriptions regardless of source.
- It was not a feature of the PDMP and EMR at the time, but it is now. It is noted in results line 152 and 153 that concurrent benzo use is 10 of 13 palliative patients.
Reviewer 2 Report
This paper describes the characteristics of patients on COTS in 2020 and what therapies had been utilized prior or in conjunction with COT. It is an interesting study and would benefit from inclusion of additional data. It is also concerning that more than 1/3 of COT prescriptions are for people with conditions such as fibromyalgia or musculoskeletal pain. More attention is needed about this issue drawing from the evidence. Comments below:
Title should potentially mention “vs intermittent use”
Introduction: Remove mention of American Pain Society which is controversial in relation to the opioid crisis (this in and of itself might be worth mentioning in the discussion).
Discussion about medical use versus non-medical use of opioids is potentially biased in favor of medical opioid use. Please provide additional data that is more balanced about this controversial issue.
Methods:
What percent of medical visits were in person versus virtual? Did opioid prescription differ based on type of visit?
How many visits were used per patient to determine COT status. What was the range of visits and patterns of use between visits? More data/details are needed.
More details about the consensus process for determining COT. What were the rates of concordance/discordance and what were reasons for discordance?
Clarify between which groups was statistical analysis run.
Results
Would like more data on duration of use of opioids and why is there an overlap between MMEs in the COT versus intermittent use groups?
It is concerning that >one third of COTs patients are for reasons such as fibromyalgia, neuropathic pain, and musculoskeletal pain. Do these patients have other medical comorbidities? The Cochrane evidence regarding use of opioid for treatment of these sorts of conditions should be described in the results.
Need more details on the use of other medications. Maybe it would be better/cleaner to look at current meds only. Otherwise, they may have been prescribed a past medication unrelated to the current pain complaint.
Need to present more data on the reasons for use in the COT group versus the intermittent group such as the methadone weaning and those using a single opioid prescription for a procedures or wisdom teeth pain.
NSAIDS have less side effects and no potential for addition. This is worth mentioning.
Author Response
Response to Reviewer 2
Thank you for taking the time to review our manuscript and for your helpful suggestions and comments. I am responding in order below with bullet points:
Title and Introduction:
- Included vs intermittent use in the title
- Removed the American Pain Society mention in the text, but left the references for those interested in that aspect of history. Did not include in discussion.
- Tried to add a little more balance in the introduction regarding misuse of opioids. I understand your point about leaning toward a medical bias.
Methods:
- Included the raw number of in person vs virtual visits in discussion line 284-5. Was unclear to me if it belonged in methods. Prescriptions were not impacted line 287.
- Visits really weren’t the determining factor. It was more the fact that patients were using monthly opioids. Our state has visits spaced 4 months apart. I did include information on yearly amounts of MME (line 128) and tried to explain the overlap a little better.
- Most of these patients were mine and getting monthly prescriptions so there wasn’t any discordance (line 286).
- Stats were run between groups in their respective tables.
Results:
- Again, 43 of the 48 patients were on opioids for all of 2020. Overlap is simply because the other 5 patients were only part of the clinic for 3 months or only started COT late in the year. If I need to make that more clear in the text, I can try.
- Addressed the FM and CRPS at the end of the discussion adding a couple paragraphs and several more references (lines 309-326)
- It would probably be cleaner to look only at current usage of medications, but that would probably require a prospective, longitudinal study. Record review looked at pain clinic records only to determine meds tried for their pain.
- I honestly don’t understand this point regarding more data on reasons for use in COT group vs IO group such as methadone weaning or procedure prescription.
- NSAID points mentioned line 232-234.
Reviewer 3 Report
Reviewer Comments:
The manuscript titled “Demographics, Diagnoses, Drugs, and Adjuvants in Patients on Chronic Opioid 2 Therapy in a Pediatric Chronic Pain Clinic” by James A. Tolley et al shows a comprehensive conclusion about the chronic opioid therapy in multidisciplinary pediatric patients. This research study has great merits and analyzed in detail in large number of patients. I appreciate authors for this research. As in current state, an average of 44 people died each day from overdoses involving prescription opioids, totaling more than 16,000 deaths. Prescription opioids were involved in nearly 24% of all opioid overdose deaths in 2020, a 16% increase in prescription opioid-involved deaths from 2019 to 2020.The number is continuously growing. Based on this study, I understood that opioid overdoses patients are equally in both sexes, which is quite interesting and sofar my understanding is that opioid undergoing treatment patients are more males than females.
The cited references are completely related to the topic and observed no typographical errors in the manuscript.
